# Working from Home, Telework, and Psychological Wellbeing? A Systematic Review

Joseph Crawford

Tasmanian School of Business and Economics, University of Tasmania, Newnham, TAS 7248, Australia; joseph.crawford@utas.edu.au

**Abstract:** The practice of telework, remote work, and working from home has grown significantly across the pandemic era (2020+). These practices offer new ways of working but come with a lack of clarity as to the role it plays in supporting the wellbeing of staff. (1) Background: The purpose of this study is to examine the current literature on wellbeing outcomes and effects of telework; (2) Methods: This study adopts a systematic literature review from 2000–2022 using the PRISMA approach and thematic analysis guided by the United Nations Sustainable Development Goals (Wellbeing, Decent Work, Gender Equality, and Inclusive Production); (3) Results: It was evident that there is a lack of clarity on the actual effects of telework on employee wellbeing, but it appeared that it had a generally positive effect on the short-term wellbeing of staff, and created more flexible and proactive work design opportunities; (4) Conclusions: There is a need for more targeted research into work designs that support wellbeing and productivity of staff, and consider the environmental sustainability changes from reduced office and onsite work and increased working from home.

**Keywords:** remote work; telework; systematic literature review; work design; workforce planning

## 1. Introduction

Telework and working from home have become necessary tools in the organization's arsenal for combatting the COVID-19 pandemic. The concept, while not a direct product of the global outbreak has moved from the periphery to the fore of work and organizing. Full time work before 2019 was typically situated in offices and workplaces onsite, with only 3.6 percent of U.S. workers and 5.4 percent of European workers required to work from home [1]. Gallup [2] finds that 37 percent of U.S. workers had engaged in some form of telecommuting within their roles, with 32 percent in 2006, and only 9 percent in 1995. In 2013, the CEO of Yahoo, Marissa Mayer, made it company policy for staff to work inside of the corporate office, and prior to 2020 this was a common position for organizations to adopt.

Indeed, studies on the nature of work, have primarily emphasized on-site work with after-hours answering of emails and international teleconferencing a secondary concern [3]. Yet, working from home is quite different than formal practices of arriving at a previously designated time, occupying a professional workspace to complete daily work, and a recommended end time for departure. Across forty in-depth interviews, teleworkers created physical, temporal, behavioral, and communicative boundaries to enable them to separate work and life [4]. Yet, the authors acknowledge that these boundaries may not be transferable to 'always on' workplaces. These boundaries orient toward regimented bureaucratic methods of organizing work [5], and offer considerable constraints to contemporary work. The resurgence of telework opportunities offers a prospective opportunity to re-evaluate restricted measures of organizing work into fixed 9-5 work hours.

This systematic review evaluates the impact of telework on worker wellbeing. This is recognizing that for workplaces—and individuals within them—to be sustainable, all workers should be supported to have and experience full and productive employment and

decent work. Decent work is Goal Eight of the United Nations Sustainable Development Goals enumerated in the historic *2030 Agenda for Sustainable Development* manifesto [6]. While worker wellbeing is only one facet of the Sustainable Development Goal, it does provide a key foundation for understanding how work effects workers. In better understanding how telework contributes to, or challenges, worker wellbeing, this review curates foundational knowledge for development of decent telework conditions. As a result, the research objective of this study is to identify the current role telework is having on worker wellbeing. To do so, the United Nations Sustainable Development Goals is used as a guiding framework to address the research objective.

### 1.1. COVID-19 Lockdown Working from Home

Many of the studies published on working from home during 2020–2022 have been based on understanding how COVID-19 has affected the wellbeing of workers. For example, Canales-Romero et al. [7] find that working from home during the pandemic did not generally contribute to negative wellbeing (although this does conflicts with much of the literature presented below). However, they did find that parents working from home who served in dual roles as 'assistant teachers' to their children did experience declining wellbeing. In this study, the emphasis was on the testing of relationships that have limited transferability to a non-COVID-19 context; working from home practices do not usually coincide with sustained home-based remote student learning of school-aged children. Instead, these are restricted to scenarios where parents choose to home school, or children have periods of school holidays, where they are not usually expected to join virtual classrooms from home.

Likewise, when reflecting on the COVID-19 stressors, it becomes clearer that working from home practices during lockdowns are materially different than working from home during or beyond the pandemic era. The development of the pandemic induced stress scale [8] assumed that living and working during lockdowns created unique stress to humans. This included the introduction of home confinement orders, economic, social, and professional loss, redesigning work practices with inadequate or uncertain resourcing, and heightened anxiety from scaled health information dissemination. This is confirmed in one study that examined how daily self-leadership enabled positive outcomes for daily basic need satisfaction during the pandemic. However, this relationship was moderated by daily rumination about COVID-19 [9]. That is, baseline thoughts about COVID-19 have a direct effect on how well individuals satisfy their own basic needs.

### 1.2. Problem Statement

There is currently inconsistent empirical evidence on how the practices of working from home and telework effect employee wellbeing. While some have attempted to resolve this, there is a need for a social and relational work design perspective on this topic. Prior to this review, there have been two reviews that have sought to respond to the relationship between telework and wellbeing. This review, however, offers a critical point of difference. The first review by Chirico et al. [10] examines 15 manuscripts on how working from home during lockdowns affected employee wellbeing published during 2020–2021. The Chirico et al. [10] review offers a useful view of how lockdown measures created conditions of declining wellbeing in employees, yet it was highly restrictive to the lockdown context. In distinguishing, this review is different from pandemic telework review, as it focuses carefully on how telework is measured, discussed, and evaluated outside of specific-pandemic lockdown scenarios. While some studies in the final sample are situated inside of the pandemic, those whose findings are related to lockdown-specific contexts were excluded.

The second review by Beckel and Fisher [1] provides short descriptions of fourteen antecedents, four mediators, six moderators, and fifteen outcome variables within a telework nomological network, grouped by updated categories originally proposed by Allen et al. [11]. Adopting a job demands-resources and macro-ergonomics systems ap-

proach, Becker and Fisher [1] examine how management-designed structures of work effect the wellbeing of employees situated in those workspaces. I acknowledge the key contribution that these scholars made to a broad-based understanding of telework structures, individual difference, and wellbeing from an occupational health perspective. In distinguishing the work of Beckel and Fisher [1] from this study, I adopt a proactive work design approach that focuses on understanding how employees and employers co-construct meaningful work environments within hybrid contexts. I focus on the telework outcome of the manager-subordinate co-construction.

## 2. Theoretical Framework: Proactive Work Design

In a 100-year review of work design research, there are five distinct historical developments identified [12]. Namely sociotechnical systems, job characteristics, job demands-control, job demand-resources, and role theory. Many of these adopt industrialist approaches to connection between work as a highly structured activity. Progression in telework and more flexible work design research have been studied in some of these contexts [1], however it seems less common that telework is examined through work designs more suited to the changing nature of work. As an outcome of this review [12], there is recognition that proactive work designs can support more positive outcomes and reduce negative outcomes in workplaces.

Grant and Parker [13] wrote on the changing nature of work and organizations because of a global transition towards a knowledge economy and away from industrial revolution notions of organizing. The study highlights proactive perspectives of work, where a higher degree of uncertainty creates a need for dynamic response. In their review, social support, outside interactions, interpersonal feedback, and social context were considered key foundations of work design theories. Interestingly, while Grant and Parker [13] focus on the work effects on individuals and organizing, Fuller et al. [14] evidence how employee proactive personality supports a response to the effect that structural and social challenges (e.g., hierarchical position, resource access) have on their felt responsibility for constructive change. That is, the extent to which people with a high proactive personality take psychological ownership of changes made. In the context of telework, employees with high proactive personalities may be more positive in responding and leading change.

While industrial and bureaucratic organizations adopt often rigid perspectives of work, where managers design jobs for employees to be placed inside of, and engage with limited agency in fulfilling those roles, it is not as common among learned employees. Bakkar et al. [9] propose that self-leadership and playful work designs enable psychological need fulfilment and role performance during the pandemic. Underlying these assumptions is self-determination, where the individual has agency to determine their work, and to achieve. In this regard, this study adopts a proactive perspective for work design, where work is built in a condition where employees can adapt their work patterns and behaviors to support their own self-determined pathway to performance. In hybrid work environments, this becomes more prominent, although there appears to be institutional resistance to self-designed methods of performing.

## 3. Materials and Methods

This study adopts a systematic review method to address the research question, using the Preferred Reporting Items for Systematic reviews and Meta-Analyses (PRISMA) Statement as guidance [15], and Braun and Clarke's [16] method of thematic analysis to define themes within the data. To ensure clarity, and that the findings in this paper reflect on telework/working from home practices, rather than outcomes that could be attributable to COVID-19 lockdowns, those studies whose primary reference point is within a lockdown are excluded. For example, where studies measure effects of government-mandates on wellbeing. Studies that were published during the pandemic, but specifically on telework and working from home and examined the wellbeing-effects of working from home were included.

### 3.1. Search Strategy

The search strategy for this study includes two phases. First, a database-driven search using Web of Science, Scopus, PsycInfo and PubMed were used. This search was limited to academic journals, English language, and date ranged from 1 January 2000 to 31 July 2022. The search phrase was:

*Title* (telework OR telecommut * OR work from home) AND *Title/abstract* (wellbeing OR well-being OR mental health OR mental ill-health).

Following the initial search, a second search using manual scanning of Google Scholar was used to identify any articles missed in the search (of the first 10 pages), with a final snowball search of reference lists for articles in the final sample. Articles identified in these stages were added at the screening stage (see Table 1 for summary).

**Table 1.** Search results.

| Database | Search 1 | Search 2 |
|---|---|---|
| Scopus | 104 results | |
| PsycInfo | 61 results | |
| Web of Science | 53 results | |
| PubMed | 34 results | |
| Google Scholar | | 18 results |
| Reference review | | 4 results |
| Subtotal | 252 results | 22 results |
| Total | | 274 results |

### 3.2. Selection Procedure and Quality Assessment

The PRISMA Statement (Figure 1 [15]) highlights the progression of 274 manuscripts identified for potential inclusion, using a review against the intention of the search strategy. That is, manuscripts must speak to telework (or the equivalent) and explicitly discuss worker wellbeing. PRISMA is commonly used in sustainability and organizational research [17]. Through a screening of title and abstracts, and a second full-text review, 43 and 70 manuscripts were excluded, respectively. The final sample was 58, and these are represented in the references with an asterisk. Importantly, considerable effort was made to remove studies that focused on the effects of COVID-19 lockdowns (*n* = 34), where the findings were not comparable to working from home contexts outside of the lockdown context. Many studies included, however, were situated within the broad pandemic landscape. The key point of difference was that these were studies not specifically on the lockdown effects, and they were not exploring relationships that were unique to the lockdown. While some conflation will exist (e.g., studies not describing when their data was collected), the aim was to only include studies that aimed to study working from home contexts.

### 3.3. Thematic Analysis

Adopting the method of thematic analysis [16], this author reviewed each manuscript multiple times during the screening process for data familiarization. Following, findings were extracted from each manuscript to support data coding and theme searching using the United Nations' Sustainable Development Goals (SDGs) as guidance [6]. Finally, each emergent theme was classified and defined, and the sample was re-assessed for inclusion to finalize each thematic representation in studies.

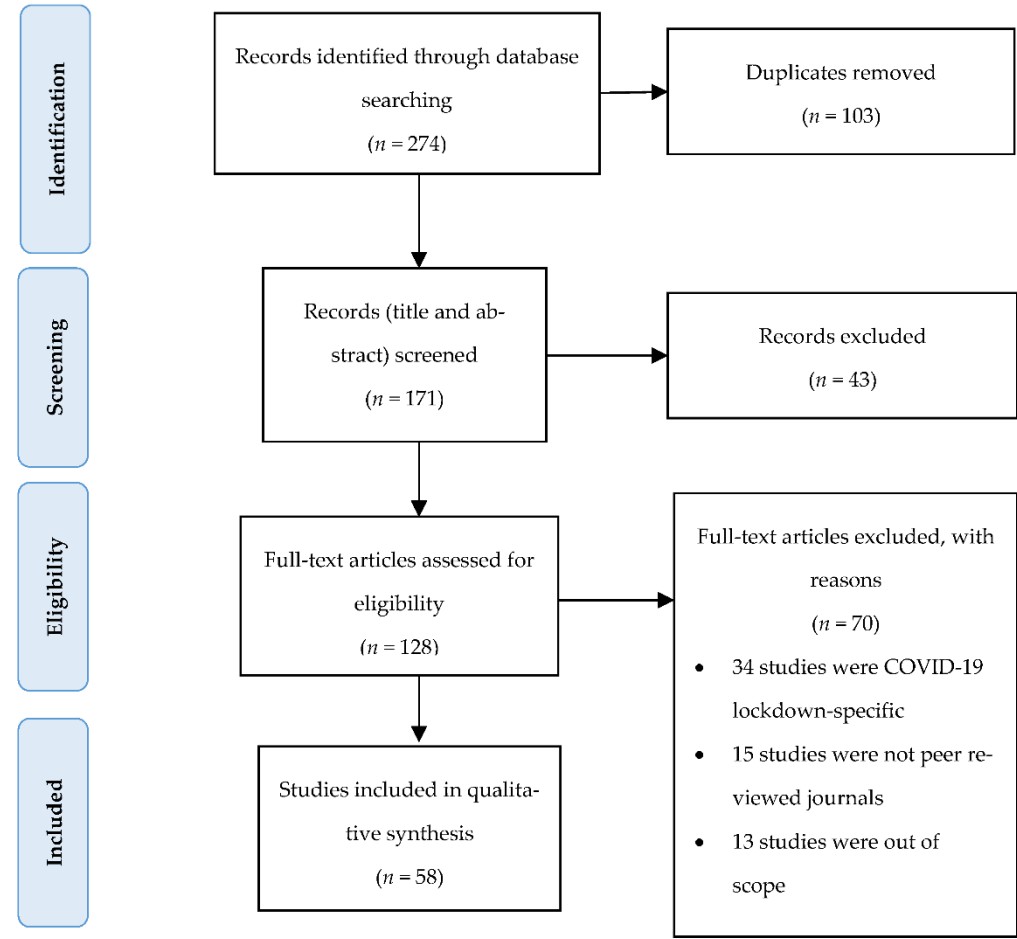

**Figure 1.** PRISMA Statement.

## 4. Results

The results of this study indicate a great deal of inconsistency among findings, with some studies reporting on conflicting relationships, and others having insignificant and significant findings on the same relationship. Greater work is clearly needed in high quality research that better controls for macro-level pandemic-differences (e.g., health information-based anxiety) in understanding how telework, in isolation, effects employee wellbeing. However, the literature appears to currently explore telework and onsite work differences from four sustainable development goal perspectives.

### 4.1. Contributing Factors for How Telework Effected Employee Wellbeing (SDG 8)

Goal 8 is to "Promote sustained, inclusive and sustainable economic growth, full and productive employment and decent work for all" [6]. Across the sample there were six broad contributing factors that explained differences in wellbeing outcomes across telework and traditional onsite work: individual differences, decent work perceptions, communication, work design, social support, and work time (see Table 2). Individual differences were generally studied, however, few studies explicitly examined how individual differences had effects on how telework was experienced. In one study, people with disability [18] experienced worse wellbeing because of telework. This is perhaps due to reduced social and physical mobility that can be experienced by some people with disability, where creating and sustaining meaningful relationships can be more difficult. In onsite environments, there are consistent relationship development opportunities (e.g., adjacent offices, shared lunch areas), whereas telework environments require planning to establish contact. Additionally, quality communication opportunities were important [19,20].

**Table 2.** How wellbeing was affected by telework.

| Variable | Source | Predicted Direction |
|---|---|---|
| **Individual differences** | | |
| Age | Arvola et al. [21] | Insignificant |
| Cultural power distance | Hoque and Bacon [18] | |
| | Michinov et al. [22] | Unclear |
| Personality | Michinov et al. [22] | Insignificant |
| Previous experience | Erro-Garces et al. [23] | + |
| | Hoque and Bacon [18] | − |
| People with disability | Adamovic [24] | + |
| Individualism perspective | Adamovic [24] | − |
| Stay-at-home children | Rieth and Hagemann [25] | − |
| Workaholism | Magnavita et al. [26] | − (moderator) |
| **Decent work perception** | | |
| Job satisfaction | Erro-Garces et al. [23] | + (mediator) |
| Motivation | Vanderstukken et al. [27] | + |
| Work meaningfulness | Maillot et al. [28] | − |
| | Maillot et al. [28] | − |
| Perceived intensity | Michinov et al. [22] | Insignificant |
| | Shepherd-Banigan et al. [29] | Insignificant |
| Work-family conflict | Vander Elst et al. [30] | − |
| Work-family balance | Miglioretti et al. [31] | + |
| Emotional exhaustion | Vander Elst et al. [30] | − |
| **Communication** | | |
| Online communication | Karatuna et al. [19] | + |
| | Kitagawa et al. [20] | + |
| Face-to-face communication | Karatuna et al. [19] | + |
| **Work design** | | |
| Academic environment | Karatuna et al. [19] | + |
| Working conditions | Karatuna et al. [19] | + |
| Office distraction | Wohrmann and Ebner [32] | − |
| Dedicated home office | Fukumura et al. [33] | + |
| | Kitagawa et al. [20] | + |
| | Fukumura et al. [33] | + |
| Flexibility | Widar et al. [34] | + |
| | Shepherd-Banigan et al. [29] | Insignificant |
| Job resources | Miglioretti et al. [31] | + |
| Job demands | Miglioretti et al. [31] | Insignificant |
| | Eguchi et al. [35] | + (moderator) |
| | Chu et al. [36] | Insignificant |
| Organizational support | Bosua et al. [37] | + |
| | Karatuna et al. [19] | + |
| Physical isolation | Wang et al. [38] | + |
| Participatory decision-making | Vander Elst et al. [30] | + |
| Autonomy | Vander Elst et al. [30] | + |
| **Social support** | | |
| Co-worker relationships | Wohrmann and Ebner [32] | − |
| Social support at home | Prabowo et al. [39] | Insignificant |
| Quality interactions | Maillot et al. [28] | + |
| Social integration | Kim et al. [40] | + |
| Psychological isolation | Wang et al. [38] | − |
| Work social support | Vander Elst et al. [30] | + |

**Table 2.** *Cont.*

| Variable | Source | Predicted Direction |
|---|---|---|
| **Work time** | | |
| Perceived time pressure | Wohrmann and Ebner [32] | + |
| After hours work | Magnavita et al. [26] | − |
| Weekend/holiday work | Song and Gao [41] | + |
| Extent/hours of telework | Heiden et al. [42] | Insignificant |
| | Vander Elst et al. [30] | Insignificant |
| Work time control | Wohrmann and Ebner [32] | + |
| Boundaryless work hours | Wohrmann and Ebner [32] | + |
| | Erro-Garces et al. [23] | + (mediator) |
| Work–life balance | Chu et al. [36] | + |
| | Zarcher et al. [32] | Insignificant |
| Work timing | Maillot et al. [28] | − |

In drawing on the proactive work design model, work design conditions were the most studied component regarding telework. For example, working conditions [19], dedicated home office spaces [20,33], flexibility [33,34], and organizational support [19,36] were seen as important contributors to decent work for employees engaging in telework practices. This makes contextual sense given that many studies focused on understanding what drivers effected wellbeing in workplaces where telework was merely a reflection of the same work and work practices occurring offsite. Likewise, general social supports from co-workers [30,32] and home family [39] were inconsistently studied and had varying results. However, when the focus was on the quality of interactions [28] and feelings of isolation [38], the sample presented a more coherent picture in line with the belongingness hypothesis. That is, it was more important for employees to have a few meaningful relationships with colleagues and family than the quantity of social relationships.

Interestingly, the area with the least congruence was arrangements of work-timing. Whereas control in work time arrangements seemed to support better wellbeing [28], the general extent of telework did not seem important [30,42]. Specific time arrangements were considered, and these had a contributory effect on the wellbeing of employees. As organizations progress towards more hybrid work arrangements, it seems greater emphasis is needed on the nature of work hours versus work results, and greater supports for developing capability for work and life task switching outside of fixed 9-5 hours of work.

*4.2. Wellbeing-Based Outcomes of Telework (SDG 3)*

Goal 3 is to "Ensure healthy lives and promote wellbeing for all at all ages" [6]. While Goal 3 focuses more on health (e.g., access to vaccines) over psychological wellbeing, it does consider promotion of positive mental wellbeing. Table 3 presents a list of wellbeing-based outcomes of telework. These include increases in positive outcomes (e.g., flow, work engagement), increases in negative outcomes (e.g., fatigue, detachment), declines in positive outcomes (e.g., sexual intercourse, emotional connections), and declines in negative outcomes (e.g., anxiety, distraction). More, however, is needed in considering and reflecting on the potential interaction effects that exist between these outcomes. This is particularly true in concepts where a measure effects telework, and in another study reports on the effect telework has on an outcome.

**Table 3.** Outcomes of telework that indicate changes in wellbeing.

| Variable | Source |
| --- | --- |
| **Increases due to telework** | |
| Self-organisation | Paridon and Hupke [43] |
| Ergonomic stress | Paridon and Hubke [43] |
| | Schade et al. [44] |
| Work engagement | Miglioretti et al. [31] |
| | Delanaeiji and Verbruggen [45] |
| Flow | Schade et al. [44] |
| Affect | Schade et al. [44] |
| Detachment | Schade et al. [44] |
| Fatigue | Oakman et al. [46] |
| Financial burden | Oakman et al. [46] |
| Onsite loneliness | Zarcher et al. [47] |
| Weight gain | Ekpanyaskul and Padungtod [48] |
| Perceived work intensity | Ekpanyaskul and Padungtod [48] |
| Next day work engagement | Darouei and Pluut [49] |
| **Decreases due to telework** | |
| Sexual intercourse | Prabowo et al. [39] |
| | Rieth and Hagemann [25] |
| | Adamovic [24] |
| Stress | Bosua et al. [37] |
| | Delanoeije and Verbruggen [45] |
| | Song and Gao [41] (increase, rather than decrease) |
| Anxiety | Schifano et al. [50] |
| Technology challenges | Liddiard [51] |
| Emotional overload | Liddiard [51] |
| Confusion | Liddiard [51] |
| Emotional connections | Liddiard [51] |
| Distraction | Zarcher et al. [47] |
| Perceived workload -Metropolitan workers -Rural workers | Turja et al. [52] (insignificant for rural) |
| Time pressure | Darouei and Pluut [49] |
| Work-family conflict | Darouei and Pluut [49] |
| Depression symptoms | Shepherd-Banigan et al. [29] |
| Physical health | Oakman et al. [46] |
| **Conflicting changes due to telework** | |
| Exhaustion | Cheng and Zhang [53] (+) |
| | Darouei and Plutt [49] (−) |
| | Windeler et al. [54] (-in part-time telework) |

Interestingly, there were some conflicting results relating to stress, with one study [41] indicating higher stress in telework, whereas four others [24,25,37,45] confirm lower stress. The study articulating higher stress [41] measured stress in 2010, 2012, and 2013, and is based on a phone survey recording the previous 24 h of activities, rather than a typical single time-point survey. Aside from this data being earlier (the studies confirming effects were published in 2020–2022), when working from home practices were less common, this study arguably provides more rigorous accounts of stress from telework practices. The difference in societal expectations and norms also may play a role. It would be reasonable to assume that in 2010–2013, telework was a less frequent work design with less resourcing established for access to the correct technology, work products, and human online connections.

Exhaustion also highlighted confused results, with positive [53], negative [49], and negative in part-time telework identified [54]. In these studies, their framing around emotional exhaustion was different. For example, one study [49] identifies morning exhaustion as an outcome of working from home, mediated by work-home conflict; whereas in another [53],

delayed emotional exhaustion was a product of extent of telework and detachment from work. These assumptions require more robust conceptualization and testing to understand how telework is actually effecting employee wellbeing, and under which conditions the best telework productivity and wellbeing outcomes are achieved.

### 4.3. Gender-Based Differences (SDG 5)

Goal 5 is to "Achieve gender equality and empower all women and girls". The literature provides some analysis on gendered differences of telework on employee wellbeing. While many of these studies do have conflicting commentary and relationships tested, some inferences can be drawn. Firstly, women tended to experience exacerbated effects of existing relationships. For example, in one study [55], women saw greater perceived advantage in telework, but also reported higher perceived disadvantage. Likewise, emotional exhaustion was considered higher in women [55], but relaxation levels were also higher in women. A key area that requires greater clarity includes stress [41], depression symptoms [29,56], and loneliness [22]. This understanding will help organizational strategists and managers develop work-based responses that may support greater gender equality. Interestingly, there were statistically significant wellbeing effects of commute time [56] and psychological distress in women [57], but not statistically significant in men. In this regard, while these studies offer a useful nomological map as to how women experience telework differently, they seem to lack the controls required to understand what differences are experienced based on gender individual differences as compared to environmental, cultural, and social differences that adversely effect people of diverse genders (Table 4).

**Table 4.** Gendered differences of telework wellbeing effects.

| Variable | Source |
|---|---|
| **Higher in women** | |
| Perceived advantages | |
| Perceived disadvantages | |
| Perceived workload | Ghislieri et al. [55] |
| Emotional exhaustion | |
| Relaxation levels | |
| Workaholism | |
| Depression symptoms | Burn et al. [56] |
| | Shepherd-Banigan et al. [29] |
| Stress | Song and Gao [41] |
| **Lower in women** | |
| Loneliness | Michinov et al. [22] |
| **Higher in women and not statistically significant in men** | |
| Commute time | Kroesen [58] |
| Psychological distress | Matthews et al. [57] |

### 4.4. How Wellbeing Was Changed by Intervention (SDG 9)

Goal 9 is to "Build resilient infrastructure, promote inclusive and sustainable industrialization and foster innovation". Most studies in the final sample tended to conduct inferential studies that linked an outcome or antecedent concept to telework (compared to onsite work). Across the findings, a series of physical structural responses supported higher wellbeing and support (e.g., stand-up desks [59]). Likewise, some supports to enable quality relationships and social support (e.g., coaching [60]; and communication strategies [24]). Having an emotionally and physically proximate dog also did well to support wellbeing in participants [59]. Yoga supported perception changes of stress, but did not present evidence of actual changes in stress or anxiety. More studies are needed in considering how experimental studies can support effective redesign of workplaces, and proactive responses to a more dynamic and flexible work environment experienced when transitioning through and between telework and onsite work (Table 5).

**Table 5.** Intervention effects on wellbeing.

| Intervention | Source | Summary |
|---|---|---|
| Communication and performance strategies | Adamovic [24] | Higher sense of belonging and wellbeing. |
| Exercise Face-to-face eye contact | Burn et al. [56] | Served as a protective factor for wellbeing declines. |
| Online behavior modification program | Falk et al. [59] | Higher affective wellbeing and performance, and lower fatigue severity. |
| Positive psychology coaching | Van Nieuwerburg et al. [60] | Higher wellbeing, reflection time, awareness, and lower negative affect. |
| Proximally and emotionally close dogs | Junça-Silva et al. [61] | Higher productivity and wellbeing. |
| Stand-up desks | Falk et al. [59] | Higher affective wellbeing and performance, and lower fatigue severity. |
| Wrist worn sensors prompting regular breaks | Zhang et al. [62] | Higher productivity and wellbeing. |
| Yoga | Wadhen and Cartwright [63] | Higher wellbeing and lower perceived stress, but no change in actual stress and anxiety. |

Management practices were also studied to understand how managers and leaders support quality responses to changing work lives among teleworkers. These often involved manager interventions to support staff through work change. For example, supporting staff to develop appropriate boundaries between work and life [64] or proactive responses to home conflict [65]. There was general coherence on the role that micromanagement has on reducing employee wellbeing. In the studies, this was represented in the negative relationship between intrusive leadership and wellbeing [26]. It was also visible when examining the relationship that supervisor trust of employees [36,40] and managers focusing more on results over specific hours worked [40] had on supporting improved wellbeing (Table 6).

**Table 6.** Leader and manager effects on employee wellbeing.

| Variable | Source | Predicted Direction |
|---|---|---|
| Boundary development | Rodrigues et al. [64] | + |
| Collaboration | Rodrigues et al. [64] | + |
| Conflict at home | Lanaj et al. [65] | + |
| Depletion | | + |
| Intrusive leadership | Magnavita et al. [26] | − |
| Isolation | Rodrigues et al. [64] | − |
| Manager results-orientation | Kim et al. [40] | + |
| | Chu et al. [36] | Insignificant |
| Supervisor trust | Bosua et al. [37] | + |
| | Kim et al. [40] | + |

## 5. Discussion

This study presented a review of the current known roles that telework has on wellbeing. While the field by age is not novel or new, in its volume of research and contemporary relevance, telework has only began to receive growing coverage in workforce planning and wellbeing research in recent years. In this study, the final sample ranged between 2006 and 2022, however, only around 14 percent of studies existed prior to 2020. Indeed, almost half the studies included were from 2022 YTD (46.5%). The emergent evidence provides useful context for scholars and practitioners, however, even with careful consideration during this review to remove studies that likely had pandemic effects included, most studies were

situated during the pandemic era. That is, some of the strength or weakness of relationships may be attributable to temporally specific rather than attributable to telework practices. In a critical assessment of the studies that were included, I now turn to consider concepts that were less present in the net of research on telework and working from home practices. I group these by work culture, environmental sustainability, and transitory states.

### 5.1. Work Culture Gaps

The studies included have begun to explore pockets of pathways by which workplaces, work designs, and people effect employee wellbeing outcomes. Additionally, this is important as scholars become clearer on what levers incentivize and hinder decent and meaningful work. Of critical importance is considering changes in the fundamental nature of work. One study wrote that when managers focus on results over specifics around hours contributed, wellbeing improves [38]. However, this could also adversely affect underserved workers or incentivize faster and lower quality work. What moderation might be needed to ensure the effect of flexible work on workers is sustainable over temporal and spatial locations.

Indeed, belongingness research has found more significant challenges to the way in which employees engage, stay, or leave workplaces. While job turnover could be considered a lagging indicator of belongingness and wellbeing [66], a better understanding of what happens to engagement, belonging, and wellbeing over periods of time is critical [67,68]. Leadership and followership co-creation practices [69] also seem to be largely absent from the telework literature. When people with specific titles (e.g., 'manager' and 'employee') come together in onsite work, the spatial conditions effect the way leadership is claimed and granted. How does the blended and hybrid spatial conditions effect how leaders and followers co-create relationships, and likewise contribute to outcomes of wellbeing and belonging?

In relation to equity and relational norms, in traditional onsite organizations, workers are afforded a degree of equivalent opportunity to network with managers, colleagues, and clients. That is, by virtue of being in the same proximate location (e.g., the work office), each employee can attempt to build relationships with most of their peers. Over time, as leader-member exchange theorists would describe [70,71], some of these relationships become psychologically close and others remain distant. For teleworkers, if they have less face-to-face interaction opportunities, will they experience heightened disconnection and social isolation from their colleagues? Likewise, equal opportunity to promotions or qualitative perceptions of their performance by managers may also be different.

### 5.2. Environmental Sustainability

Among the studies, many examined social and physical changes experienced by teleworkers in contrast to similar onsite workers. Yet, there were few studies that discussed the impact of decentralized work structures on environmental outcomes. This seems congruent with work on higher education during the pandemic, that indicated environmental sustainability was deprioritized in the place of continuity of work and learning. In one study, working from home was identified as reducing transport costs [72]. However, in pandemic studies that feature working from home, eating habits were seen to be healthier [73], yet it was not clear if out-of-home eating changed. Higher consumption of takeaway food from cafés and restaurants can have a contributory effect on landfill and single-use plastic consumption, when contrasted to home-based meal preparation.

In considering electricity consumption, COVID-19 mandates that effected work from home patterns saw increased power consumption at home by 13 percent [74]. It is however, unclear if the increased domestic electricity consumption features a decline in office and work environment levels by an equivalent level (i.e., less or more overall electricity consumption). There is more research needed with relation to the relative effects of social and environmental outcomes and differences in telework contexts, including controlling for pandemic-effects.

*5.3. Effects beyond Transitory States*

Change creates inertia, and change creates resistance and differences in affective experiences. For many of the studies in this sample, scholars produced pilot interventions of introducing telework conditions or measuring the differences between employees who were teleworking and those completing onsite work. Yet, in the latter, it was rarely clear if these teleworkers were only recently transitioned to this type of work or if studies were capturing genuine teleworkers in contrast to genuine onsite workers. Studies pertaining to telework moving forwards should provide clear parameters for the previous experience and temporal duration of telework experienced by those sampled.

To extend, during the pandemic, there are numerous studies on changing wellbeing because of lockdowns [75,76]. These changes are likely having an exogenous effect on employee wellbeing that is exacerbating the effects theorized as endogenous of telework. As the world moves through and beyond the pandemic, some of the assumptions highlighted within this research will need to be re-tested and better control for exogeneity in the telework and onsite work experience.

*5.4. Practical Implications*

This study focused on examining the current research on telework work conditions in contrast to onsite work. I was primarily concerned telewith establishing a clear understanding of the published literature on telework and employee psychological wellbeing to support future research [77–81]. However, this study has clear implications for practitioners experiencing, or implementing telework. First, telework is not comparable to onsite work. While it seems it contributes to better employee wellbeing outcomes, the reasons why this is the case are more mixed and often conflicting. This means that focusing on creating an environment that works for the specific industry and individual needs is important, and the variability in relationships tested in the sample studies may be reflective of the complexity of employees and specific work practices. Second, telework removes physical boundaries that separate work (in the office/onsite) and life (not in the office/offsite), and this requires a resetting practice for workers. Working from home, productivity, and employee health remain linked [82–86]. While managers could support employees to build effective boundaries, there may be a case for progression towards results-based evaluation rather than performative hours-based work; particularly in knowledge workers.

Third, identifying strategies to build connection and social cohesion between onsite, telework, and hybrid staff is critical for ensuring that work modality does not affect long-term performance, wellbeing, engagement, or belongingness among staff. This could include practices such as mandated onsite days, although this likely offers a disadvantage those who are required to transition to onsite on some days without social and physical systems (e.g., childcare and parking permits) in place. Higher education have been studying students transitioning between modalities for a while [86–90], and could be drawn on in the context of working from home. Indeed, these staff may also find it difficult to reintegrate with the social bonds developed by permanently onsite staff also [91]. It may be more effective to choose neutral easy-access locations for regular blended social and professional meetings and check-ins. Local parks may offer an interesting contribution. In the Australian small business context, attending a public barbeque in the park for a lunch meeting could offer a useful opportunity for developing meaningful connections. Being physically co-located (e.g., shared or adjacent offices) may support social bonding, but without sustained opportunities to connect, this may be more complex.

## 6. Conclusions

This study examined the relationship that telework and working from home practices had on employee wellbeing. Through a systematic review, leveraging PRISMA and thematic analysis, I evaluated 58 studies on this relationship using the United Nations' Sustainable Development Goals as guidance. While the evidence was not always clear, it did seem to highlight that opportunities for telework seemed to generally improve the

wellbeing of employees. However, this was not a linear relationship with extent of telework not always supporting heightened wellbeing. The data indicated that telework was different for some staff, and that telework could be considered decent work in parallel or substitution of onsite work. The research on telework however was often conflicting, and likely a product of conflation between lockdown-effects from the COVID-19 pandemic and genuine telework. While effort was made to control for these in the sample, the majority of studies were published between 2020–2022. This is an important limitation to the work. There are incredible opportunities for telework to create more meaningful, flexible, and productive work environments where employees can belong across spatial contexts. However, it requires dedicated and considered management and leadership to support staff to transition and stabilize such a change.

**Funding:** This research received no external funding.

**Institutional Review Board Statement:** Not applicable.

**Data Availability Statement:** Not applicable.

**Conflicts of Interest:** The authors declare no conflict of interest.

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
