# Peer review of "Working from Home, Telework, and Psychological Wellbeing? A Systematic Review"

_sustainability, doi:10.3390/su141911874_

Round 1
Reviewer 1 Report
In this manuscript, the author reports the results of a systematic review of the well-being effects of telework. Such a literature review was definitely needed, due to the increased relevance of working from home/telework as a consequence of the Covid-19 pandemic, and the ongoing debate about its consequences for workers’ wellbeing and productivity.
The author did a good job in following the PRISMA approach to identify the most relevant articles from 2000 to 2022 and summarized the main results following the UN SDGs. There are, however, a couple of issues which I encourage the author to address.
First, since the aim of the review was to identify general outcomes of telework (and not outcomes specifically linked to working from home during the Covid-19 lockdowns), the author states that “studies whose primary reference point is within a lock-down are excluded.” (page 3). The vast majority of studies included in the review, however, have been conducted during the lockdowns (“…only around 14 percent of studies existed prior to 2020. Indeed, almost half the studies included were from 2022 YTD (46.5%).”, page 10), therefore it is almost impossible to disentangle the effects of lockdown-related issues (e.g., kids are not a school while you’re working from home) from the general effects of telework on workers’ wellbeing. I suggest removing (or toning down) the aim of finding results not influenced by the recent lockdowns.
My second issue is about the relevance of the findings. In sum, it seems that telework has a generally positive effect on workers’ well-being, but there is also a general lack of clarity since there is a great deal of inconsistency among findings. The reader, however, is left to wonder why there are such conflicting results in the literature since there are no in-depth analyses or explanations. For example, there are conflicting results also on stress, one of the most important outcomes of working from home, and the author writes “with one study [39] indicating higher stress in telework, whereas four others [22-23, 35, 44] confirm lower stress” (page 8), but that's all. Why that one study found different results? Are there some differences among the studies that could explain this inconsistency? I believe that a literature review, to be really useful, should at least try to explain the most relevant or surprising inconsistencies.
Lastly, another round of proofreading is needed (e.g., “Bakkar et al. [9]” instead of “Bakker et al. [9], page 3; in Table 2 the variable “Quality interactions” should start on a new line; Table 3 is named Table 2 on page 7).
Author Response
Dear Reviewer,
Thank you for the feedback. I have uploaded the revision letter below, and I am uploading a manuscript with track changes.
Kind regards,
Author

Reviewer 2 Report
Comments to the Author
Recommendation: Moderate revision
First, I would like to thank the Editor for trusting me with the opportunity to review this research, "Working from home, telework, and psychological wellbeing? A systematic review.”
The theme of the study is timely and exciting; hence has the potential to contribute to the literature on Telecommting/work from home subject to a moderate revision. Some of the comments/suggestions are given as follows:
1- Although in the study, the author has referred to two review studies on telecommuting and its role on psychological wellbeing and based this study accordingly, I still feel that the Introduction section could have been more problem-specific while building upon the research gap(s). The introduction lacks the specificity of the objectives of the study.
2- The theory section seems too short and could have been lengthened.
3- The study is a systematic literature review following PRISMA protocol using search query Title (telework OR telecommute OR work from home) AND Title/abstract (wellbeing OR well-being OR mental health OR mental ill-health, I feel that the author could have used more keywords for telecommuting and wellbeing as there are a bunch of keywords being in use interchangeably. The use of more keywords could have fetched more studies.
4- Inclusion/exclusion criteria of a manuscript into the study are not defined clearly and in a detailed manner.
5- The results and discussion sections seem fine.
6- Moderate language editing and proofreading are also required.

Author Response

(The authors gave the same response as above.)
